# Path loss measurement and modeling of 5G network in emergency indoor stairwell at 3.7 and 28 GHz

Md Abdus Samad[1], Dong-You Choi[2], Kwonhue Choi[1]*

1 Yeungnam University, Gyeongsan-si, Gyeongsangbuk-do, South Korea, 2 Chosun University, Dong-gu, Gwangju, Jeollanam-do, South Korea

* gonew@yu.ac.kr

## Abstract

Research on path loss in indoor stairwells for 5G networks is currently insufficient. However, the study of path loss in indoor staircases is essential for managing network traffic quality under typical and emergency conditions and for localization purpose. This study investigated radio propagation on a staircase where a wall separated the stairs from free space. A horn and an omnidirectional antenna were used to determine path loss. The measured path loss evaluated the close-in-free-space reference distance, alpha-beta model, close-in-free-space reference distance with frequency weighting, and alpha-beta-gamma model. These four models exhibited good compatibility with the measured average path loss. However, comparing the path loss distributions of the projected models revealed that the alpha-beta model exhibited 1.29 dB and 6.48 dB for respectively, at 3.7 GHz and 28 GHz bands. Furthermore, the path loss standard deviations obtained in this study were smaller than those reported in previous studies.

## Introduction

The indoor facility-based wave propagation model has garnered attention owing to the ongoing development of high-quality wireless communication services [1]. Modeling the wave propagation can aid in planning and managing wireless communication links to provide high-quality network services for wireless communication systems [2]. Wireless propagation has several applications, such as finding objects in indoor environments, as conventional global position system does not perform well in indoor environments. For example, to find a target object in an indoor environment has a wide range of possible applications, including security, emergency services, health care, and commercial areas [3], which can be used for be determined based on the received signal strength in indoor areas. However, due to the complicated multipath propagation within structures, it is challenging to offer precise positions via received signal strength indicator [4]. In addition, wireless propagation has several application services, such as rescue or military operation [5], providing quality networks to indoor environments. Consequently, researchers continuously try to enhance the existing wave propagation techniques in indoor environments [6, 7].

**Funding:** This work was supported by the National Research Foundation of Korea (NRF) grant funded

by the Korean government (MSIT) under Grant 2021R1A2C1010370. There was no additional external funding received for this study. The funders had no role in study design, data collection and analysis, decision to publish, or preparation of the manuscript.

**Competing interests:** The authors have declared that no competing interests exist.

In the literature, different types of indoor environments such as corridor [8], hall [9], office [10], laboratory [11], indoor production site [12], and library [13] were proposed.

However, although stairs are necessary for everyday use, particularly in emergencies, research on wave propagation in stairs is limited compared to other indoor spaces. The limitations include the number of research findings and the different modeling approaches used in indoor stairs compared to other indoor environments.

In [14], radio wave propagation was measured from 1.2 to 1.8 GHz on a staircase connecting many floors. Then the authors proposed a propagation model based on a three-dimensional ray-polygon tracing model was proposed.

The distance between the transmitter and the receiver in three-dimensional space is crucial to determining the path loss for a specific radio link. Because of the importance of the separation distance, most path loss models assume that the path loss increases proportionally to the square of the separation distance. A term referred to as the path loss exponent factor (PLE) is often used to realize this value of the exponent factor by two. However, in the case of the multipath effect, the proportionality factor (e.g., path loss exponent rule) must be adjusted, which results in different values of PLE n, depending on the multipath phenomena. Path loss is a linear function that indicates the path gain decrease (or increase) behavior when expressed on the decibel scale. It follows a logarithmic distribution around the values expected from the mean path loss [15]. Because stairs and multistory buildings are ubiquitous in modern cities, it is crucial to understand the working of different wireless devices on stairs.

To ensure safety, the uninterrupted use of wireless communication links is essential in emergency areas. In high-rise buildings, for instance, rescue operations [5] or military activities during combat rely on having reliable network connectivity through an emergency exit. Besides being essential for emergencies, stairs are necessary to move from one floor to another. As a result, it is critical to developing a suitable path loss model for wireless communication on emergency stairs.

An image-based ray-tracing investigation of radio propagation measurements in a multi-story staircase reported the minimal effect of reflections on the walls of the stairs [16–18]. In [15], the authors investigated path loss modeling and extracted the statistical parameters for four typical staircases with vertical and horizontal polarizations. The path loss model based on the accumulation distance (the walking distance between the transmitter and receiver) was more accurate than that based on the standard three-dimensional separation distance between the transmitting and receiving antennas. In [19], a path loss model was demonstrated for stairwell environments using observations made at 900 and 1800 MHz in four stairwells. A deterministic ray tracing model was shown for an indoor staircase channel at 10 GHz. Further, heuristic diffraction coefficients were used to determine the impact of diffraction from the edges of stairs in [20]. In [21, 22], the propagation of the mmWave band was tested and compared on stairs for various polarization combinations. In [23], the channel properties of the staircases were analyzed and compared in centimeter and mmWave wavebands.

In [24], the path loss was found to have a lower slope on the far-end floor than on the near-side floors of the receiver. According to [15], typical values of the path loss exponent for distinctive indoor and outdoor propagation settings vary from 2 for open spaces to between 3 and 5 for shaded urban cell radios and between 4 and 6 for blocked buildings. [24] reported that the typical path loss exponent values for various locations in various vertical construction facilities were stated at different levels in the indoor environment. The path loss exponent was 2–6 when considering propagation on the same floors and across other floors. In multistory buildings, a regular increase in the path loss exponent values was observed as the signal moved through more floors. However, despite these several studies, further measurements are required to investigate the propagation occurring inside different stairs of various shapes and

purposes. Various physical phenomena, such as diffraction, reflection, refraction, and scattering, can occur in a propagated wave. Owing to these phenomena, polarization deviations can occur during the propagation of radio waves on indoor stairs. Therefore, multipath effects are more noticeable inside the stairs in contrast to outdoor propagation.

This study is the continuation of our previous study of path loss measurement [8, 9, 25] in closed indoor environments. In this study, we investigated a particular type of stairwell, a bent stairwell with a wall partition, at the campus of Chosun University in Gwangju, South Korea.

We conducted measurements at frequencies of 3.7 and 28 GHz using different antenna configurations. We selected 3.7 GHz as the frequency currently deployed for the 5G network in many countries [26], and the 28 GHz frequency as the next probable operating frequency [27]. The highlights of this study can be summarized as follows.

- Path losses were measured up to the fifth floor of an academic building, and consequently, the average path losses in the measurement positions along the vertical structure were determined. A horn and an omnidirectional antenna were used to compare their effects on path loss.

- The average received power level was used to determine the optimized environment and frequency-dependent specifications of large-scale techniques, for example, close-in, floating intercept, close-in with frequency-weighted loss component, and alpha-beta-gamma.

The remainder of this paper is organized as follows. Section 2 describes the experimental stairs considered in an academic building. Section 3 introduces experiment-specific situations and explains their associated parameters. Section 4 presents the mathematical background of the large-scale models considered. Further, visual representations of the simulated large-scale path loss models computed using the statistical parameters of path losses and measured data are presented in Section 5. Finally, Section 6, concludes the study conclusions.

## Stair structure

We examined the stairs of a bent stairwell with wall partition at the main academic buildings of Chosun University in Gwangju. The stairs are located in the south-facing location of the main academic building of the university. A cross-sectional view of the stairs is shown in Fig 1.

The path loss model for the indoor stairs produced in this study is valuable and applicable to many comparable stairs owing to these stairs being standard and available at many contemporary workplaces and structures. Although the materials on the steps can affect the measurements, their effects are less critical than those of the walls, stairs, and ceilings of the stairs, which reflect and transmit the maximum light. Fig 2(a) shows a photograph of a measurement setting where the receiver is located on the second floor, whereas Fig 2(b) shows a detailed view of the Rx system.

According to our observations, most staircases are constructed using reinforced concrete, with their walls typically made of gypsum or concrete panels. In contemporary workplaces and structures, ceilings and floors are typically constructed using concrete. The commonality in the construction and materials of stairs in modern office buildings facilitates the application of the results obtained in this study to most, if not all, indoor stairwell propagation environments in modern facilities with several stairs. The height of the academic building was up to the fifth floor at the experimental stairwell site. The height and width of the step were 13 and 15 cm, respectively. We set the receiver at 2.5 m from the end of the stairwell to match the end of the dividing wall between the stairs. Further, the outer side of the building shared the boundary wall of the building, which had several glass windows that were closed during the experiment. The stairs from the first to the third floor are of the closed type (owing to the presence of doors to

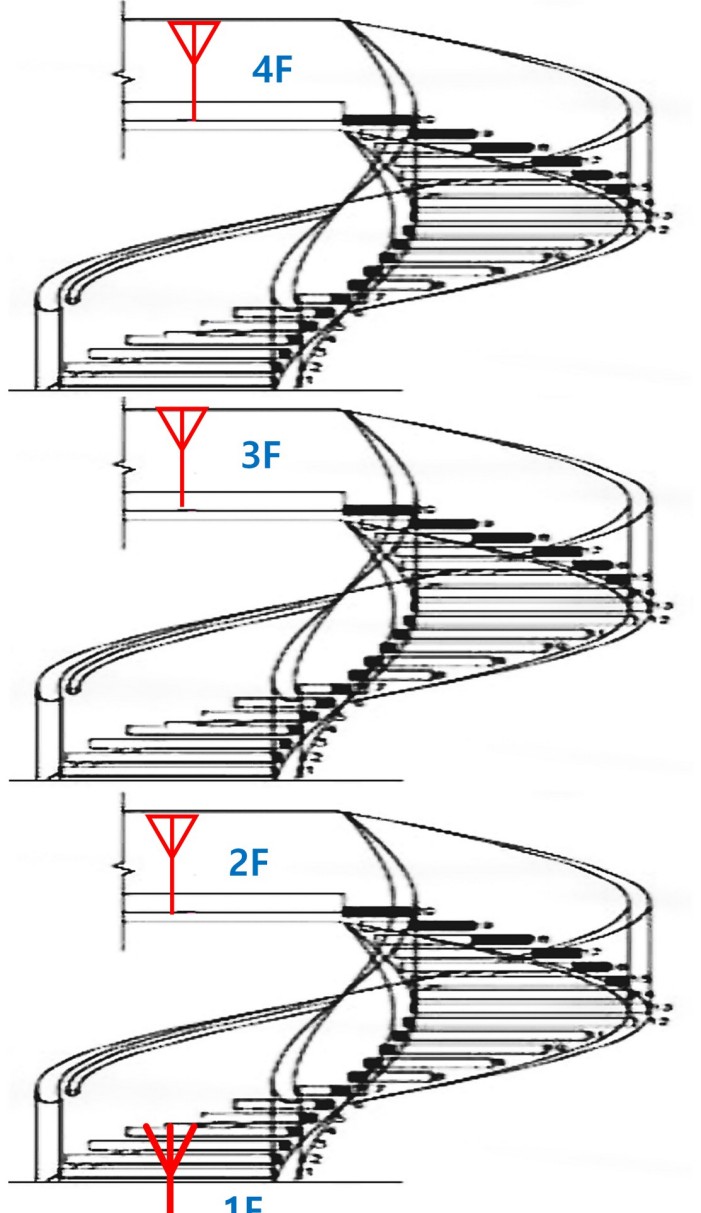

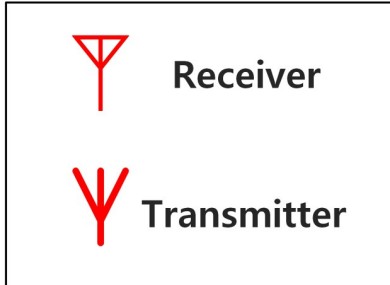

**Fig 1. Partial structure of emergency exit stair.**

block the use of the stairs to enter these floors, Fig 3(a)). During the measurement, all additional blockage door facilities were closed. However, there was no temporary closure facility from the fourth to the fifth floor (Fig 3(b)). Measurements were conducted with all temporary closure facilities open. In addition, there were open corridor spaces at the beginning and end of the stair. Consequently, no actions were undertaken, such as closing or opening doors or windows.

## Measurement systems and data collection procedure

We collected the measurement data using the same data collection setup described in [28], with minor adjustments to the antenna settings. A low-noise amplifier with a gain of 57 dB was added to the receiver's front end to extend the dynamic measurement range. At 3.7 GHz

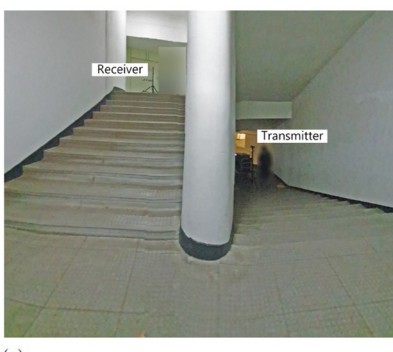
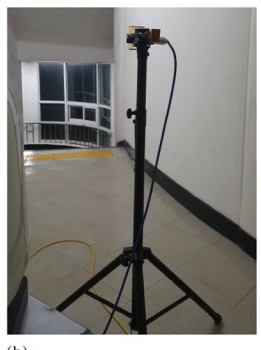

**Fig 2. Measurement setting snapshot.** (a) The Tx and Rx were together, where the Tx was on the first floor, and the Rx was on the second floor. (b) Detailed location of Rx on the second floor.

operational frequency, we used two types of antenna systems: omnidirectional (ODA0467–10) and horn (BWA0218–10) antennas, whereas at 28 GHz, only a horn antenna (WR28–20A) was used. TA Engineering Inc., South Korea, manufactured all three antennas. Three types of links were arranged at each experimental site. The same 3.7 GHz horn antenna was used at the Tx and Rx sites, with a co-polarization configuration and the same horizontal-horizontal polarization configuration. In another link, a 3.7 GHz horn antenna was used at the Tx site, whereas a 3.7 GHz omnidirectional antenna was used at the Rx site. Furthermore, in the last link, we used a 28 GHz horn antenna at the Tx site and a 28 GHz horn antenna at the Rx site. The horizontal beam width at the 3.7 GHz antenna ranged from 40˚ to 45˚, whereas at the 28 GHz antenna, it ranged from 40˚ to 45˚. Fig 4 shows the block diagram of the measurement system. The 3.7 GHz omnidirectional antenna recordings merely provided a background for the historical context of the measurement missions.

The Tx horn antenna was mounted on a metallic tripod at the height of 1.75 m. Similarly, the Rx horn antenna was installed on a metallic tripod; however, the Rx omnidirectional antenna was specially installed for experimental purposes and consisted of a metallic table-like structure with four wheels. The beam widths of the 3.7 and 28 GHz horn antennas were 40˚ and 18˚, respectively, with horizontal polarization at the 3 dB point. The bandwidth of the channels in all measurements was 200 MHz. Further, the Tx antenna was fixed at ground level at the experimental site. Furthermore, the Tx antenna was tilted by + 15˚, whereas the Rx

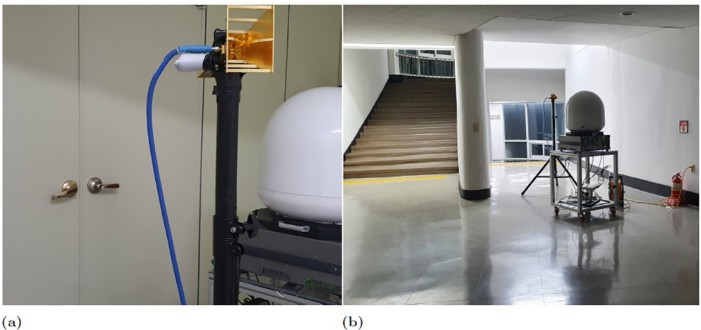

**Fig 3. Measurement setting snapshot.** (a) The temporary door facility on the first, second and third floors. (b) There was no such temporary door facility on the fourth and fifth floors.

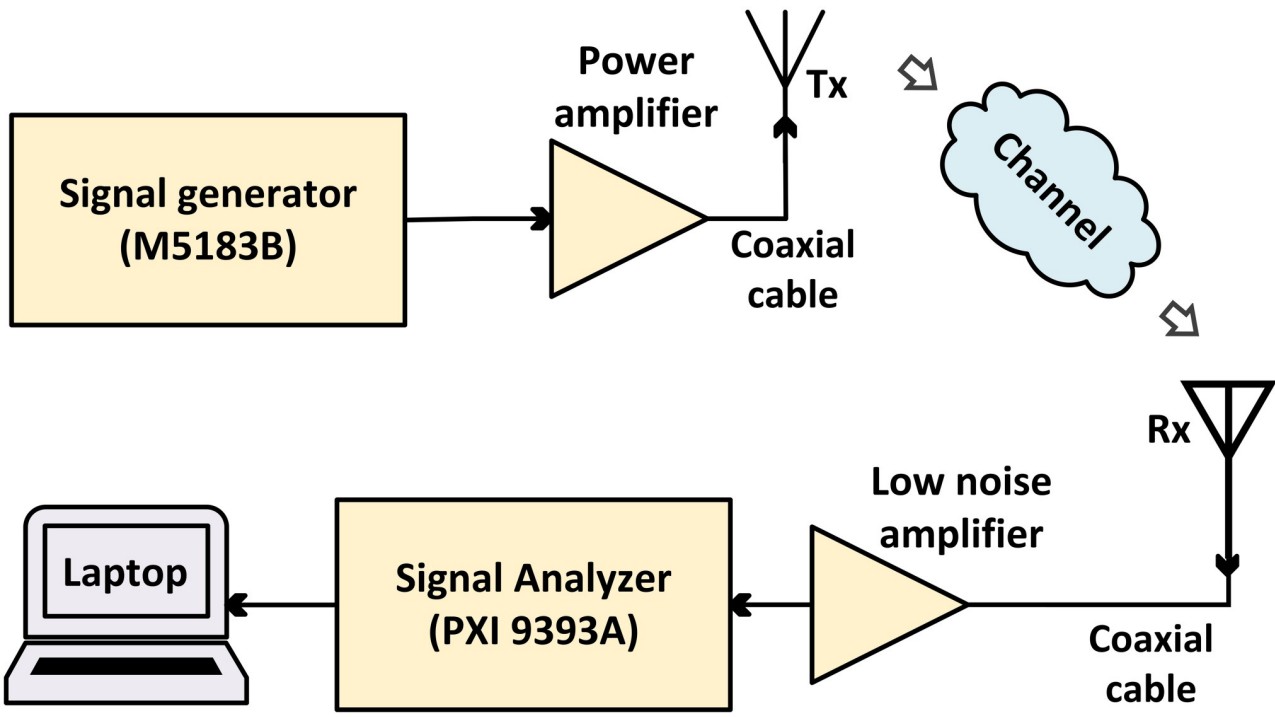

**Fig 4. Simplified channel sounder.**

antennas were not inclined. The path loss was not generated directly by the measuring system. The measurement system's vector signal analyzer gave us the received signal strength. The path loss was calculated by taking into account the gain and attenuation characteristics of the transmission line at the source and the sink terminal. Since all the loss or gain was expressed in decibels, we added up all the gains and subtracted out all the attenuations in the channel other than the wireless transmission path to get the observed path loss. Therefore, (1) was used to determine the power loss (P) in the wireless transmission link:

$$P = (P_t + G_{at} + G_{ar}) - (P_r + C_{\text{loss},t} + C_{\text{loss},r}) \tag{1}$$

where $P_t$ is the transmitted signal strength level, $G_{at}$ is the transmitter antenna gain in decibel, $G_{ar}$ is the receiver antenna gain in decibel, $P_r$ is the received signal level at the receiver side, $C_{loss,t}$ is the amount of attenuation in the transmission cable at the transmitter in decibel, and $C_{loss,r}$ is the amount of attenuation in the transmission cable at the receiver side, in decibel. Tables 1–3 show the power levels of the transmission system in different sections and the calculated path loss determined using (1).

**Table 1. Measurement data at 3.7 GHz, Horn-horn antenna.**

| TX | Power (dB) | Location | Received power (dB) |
|---|---|---|---|
| SG Power | 20 | 2F | -62.6318 |
| Cable Loss | -2.8 | 3F | -85.06 |
| Ant Gain | 10 | 4F | -94.7345 |
| Net Power | 27.2 | 5F | -109.512 |

**Table 2. Measurement data at 3.7 GHz, Horn-omnidirectional antenna.**

| TX | Power (dB) | Location | Received power (dB) |
|---|---|---|---|
| SG Power | 20 | 2F | -67.38909091 |
| Cable Loss | -2.8 | 3F | -85.01727273 |
| Ant Gain | 10 | 4F | -99.69909091 |
| Net Power | 27.2 | 5F | -106.7190909 |

## Path loss analysis of wave propagation

In recent years, many researchers and organizations have proposed environment specific path loss models. Consequently, many researchers have used the logarithmic distance path loss model for indoor environments. The logarithmic distance path loss model is formulated as [29]:

$$P(d) = \text{PL}(d_0) + 10n \log_{10}\left(d/d_0\right) + \xi, \tag{2}$$

where $n$ is the path loss exponent, $P(d)$ denotes the path loss in decibels at a distance of $d$ from the transmitter, $\text{PL}(d_0)$ denotes the path loss in the reference separation span from transmitter $d_0$, and $\xi$ is a random variable with a zero-mean Gaussian distribution. The finite standard deviation of $\sigma$ in the Gaussian distribution represents the amount of shadowing [29].

Considering the effect of frequency on path loss (2), the formula can be expressed as follows:

$$P_{\text{CIDF}}(f, d)[\text{dB}] = \text{FSPL}(f, d_0) + 10n \log_{10}(d) + \xi_{\text{CIDF}}, \tag{3}$$

where $P_{\text{CIDF}}$ is the path loss dependent on the parameters of distance $d$ and frequency $f$ in the CIDF path loss model. The symbol $\xi_{\text{CIDF}}$ is a Gaussian random variable with mean and standard deviation of zero and $\sigma$, respectively. Further, $\text{FSPL}(f, d_0)$ is the projected path loss at a distance of $d_0$ from the transmitter and can be expressed as $10 \log_{10}\left(\frac{4\pi d_0}{\lambda}\right)^2$, where $\lambda$ is the wavelength of the carrier frequency, and the symbol $n$ is the PLE. The optimized PLE and shadowing factor distributions were calculated using the procedure described in [28].

The ABM path loss models use a constant $\alpha$ as the floating intercept parameter on the path loss axis, and $\beta$ is the slope of the path loss line. The ABM model can be represented as [30]:

$$P_{\text{ABM}}(d)[\text{dB}] = \alpha + 10 \cdot \beta \log_{10}(d) + \xi_{\text{ABM}}, \tag{4}$$

where $P_{\text{ABM}}$ is the path loss dependent on the distance parameters $d$, $\alpha$ is the intercepting parameter in unit $dB$, $\beta$ is the slope of the line, and $\xi_{\text{ABM}}$ is a Gaussian random variable with mean and standard deviation of zero and $\sigma$, respectively.

Although the parameter $\text{FSPL}(f, d_0)$ in (4) appears similar to the parameter $\alpha$ in (3), it is different. The difference is that the parameter $\text{FSPL}(f, d_0)$ in the CIDF model is physically

**Table 3. Measurement data at 28 GHz, Horn-horn antenna.**

| TX | Power (dB) | Location | Received power (dB) |
|---|---|---|---|
| SG Power | 20 | 2F | -81.54545455 |
| Cable Loss | -9.4 | 3F | -114.8230769 |
| Ant Gain | 20 | 4F | -108.2653333 |
| Net Power | 30.6 | 5F | -121.9992857 |

significant, whereas the parameter in $\alpha$ is not. However, although they lack similarity, many authors consider these parameters to be the equivalent as (FSPL($f$, $d_0$) to $\alpha$ and $n$ to $\beta$) and compared the parametric statistics of these two models together [31]. Therefore, in this study, we considered the intercept parameter ($\alpha$) to be equivalent to the loss of the free-space path, and the slope ($\beta$) was equivalent to the PLE. The optimized values of $\alpha$, $\beta$, and shadow factor were calculated using the procedure described in [28].

The effects of frequency on the path loss were ignored after a reference distance (say, $d_0 = 1m$) in both the CIDF and ABM models. However, in [32], the CIDF model was transformed into the CIDMF model by considering the effect of frequency on path loss after the reference distance $d_0$.

A frequency-dependent path loss exponent model (CIDF) can be transformed into a CIDMF model with certain modifications to the original CIDF model. The CIDF and CIDMF models employ the equal physical importance of the loss of the free-space path at the radius of the reference distance for the same reason. Thus, the path loss equation of the CIDMF model can be calculated as [32]:

$$P_{\mathrm{CIDMF}}(f, d)[\mathrm{dB}] = \mathrm{FSPL}(f, d_0) + \left( n(1 - b) + nb \cdot \left( {f}/{f_0} \right) \right) \cdot 10 \cdot \log\left( {d}/{d_0} \right) + \xi_{\mathrm{CIDMF}}, \quad (5)$$

where $P_{\mathrm{CIDMF}}$ is the path loss dependent on the parameters of distance $d$ and frequency $f$ in the CIDMF model. Further, the term $n$ represents the distance dependence of the path loss exponent, and $b$ is an optimization specification that defines the linear dependence of the path loss on the weighted average of the frequencies $f_0$ (in GHz). In addition, $\xi_{\mathrm{CIDMF}}$ is a Gaussian random variable with a mean and standard deviation of zero and a finite value $\sigma$, respectively.

Alpha-beta-gamma (ABGM) is a path loss propagation model applicable to all generic frequencies on a large scale. However, the model is not suitable for specific application areas and is only helpful for specific scenarios such as LOS and NLOS radio links. The path loss, when using the ABGM models, is calculated as follows [32]:

$$P_{\mathrm{ABGM}}(d, f)[\mathrm{dB}] = 10\alpha \log_{10}(d) + \beta + 10\gamma \log_{10}(f) + \xi_{\mathrm{ABGM}}, \quad (6)$$

where $P_{\mathrm{ABGM}}$ is the path loss dependent on the distance parameter $d$ and the frequency $f$, with a reference distance $d_0$ of 1 m. The functional parameters result in decibels when distance and frequency are measured in meters and gigahertz (GHz), respectively. Further, the symbol $f$ refers to the carrier frequency, where $d$ is the three-dimensional Euclidean distance between the transmitter and receiver. In addition, coefficients $\alpha$, $\beta$, and $\gamma$ are the statistical parameters of the ABGM model. The logarithmic distance and logarithmic frequency coefficients are the parameters $\alpha$ and $\gamma$, respectively, and $\beta$ is the offset parameter used to determine a better path loss. The random variable $\xi$ models a substantial variation in the signal along the radio-link transmissions. Typically, the noise distribution $\xi_{\mathrm{ABGM}}$ is assumed to follow a Gaussian distribution with a zero mean, and a finite variance $\sigma^2$ is required.

The minimum mean square error optimization technique can be used to determine the optimal values of the factors $\alpha$, $\beta$, and $\gamma$ according to the procedure in [28]. The ABGM method was applied to a particular dataset via the calculation of the optimized values of the parameters $\alpha$, $\beta$, and $\gamma$.

## Path loss prediction and discussions

As mentioned, sophisticated tools have been used to measure the strength of the signal received in NLOS situations. In this investigation, we used the large-scale path loss models CIDF and ABM for single-frequency operation and the CIDMF and ABGM models for

**Table 4. Mean and standard deviation of the measured received power.**

| Floor | Mean, standard deviation | 3.7 Horn (dB) | 3.7 Omnidirectional (dB) | 28 Horn (dB) |
|---|---|---|---|---|
| 2F | Mean | -62.6277 | -67.376 | -81.5263 |
|  | Standard deviation | 0.00490292 | 0.139846 | 0.00174108 |
| 3F | Mean | -85.0795 | -84.0965 | -114.779 |
|  | Standard deviation | 0.0357964 | 0.169044 | 0.77384 |
| 4F | Mean | -94.7187 | -99.8026 | -108.096 |
|  | Standard deviation | 0.0553958 | 1.77112 | 0.253981 |
| 5F | Mean | -109.481 | -106.784 | -121.94 |
|  | Standard deviation | 0.520163 | 2.42514 | 2.00226 |

multiple-frequency operation. Using various antennas, we set up three distinct types of links at frequencies of 3.7 GHz (using a horn antenna), 3.7 GHz (using an omnidirectional antenna), and 28 GHz (using a horn antenna).

All the considered frequency units in this study are in the GHz band. Hence, in case of a missing frequency unit, it will be in the GHz band. Referring to all considered models, we used a reference distance ($d_0$) of 1 m. In addition, we considered H-H for the horizontal-to-horizontal radio link from the transmitter (Tx) to the receiver (Rx) link. Table 4 lists the mean and standard deviation of the datasets measured at different points and measurements. Furthermore, Figs 5–7 show the variations in the measured datasets, which were fitted to the Gaussian distribution $\xi(\mu, \sigma)$. As evident, when the receiver moved far away from the transmitter, the received power exhibited a wider Gaussian distribution.

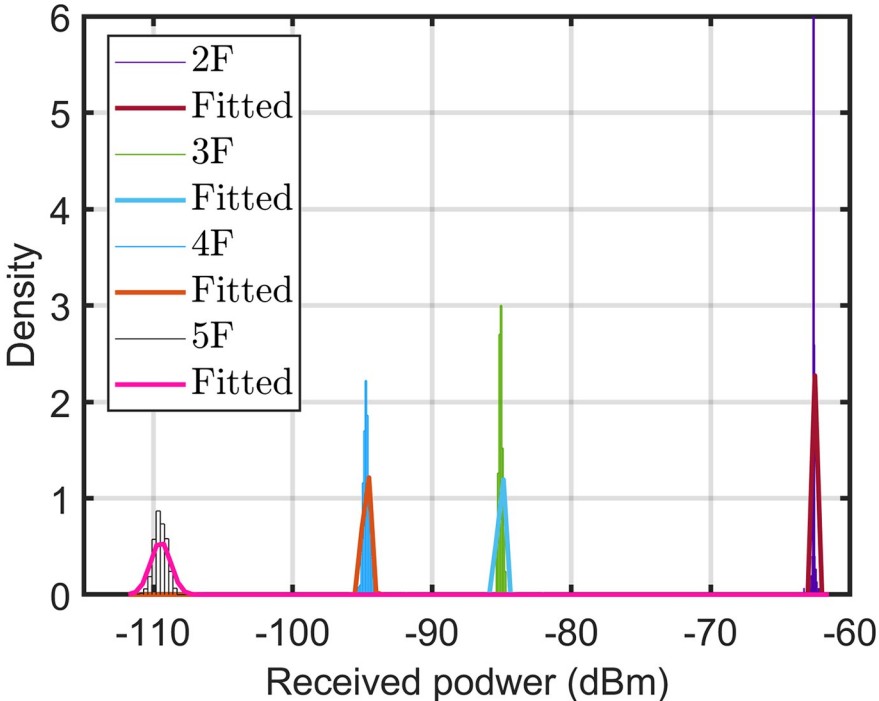

**Fig 5. Distribution of measured path loss along vertical Euclidean distance at 3.7 GHz using the horn-horn antenna.**

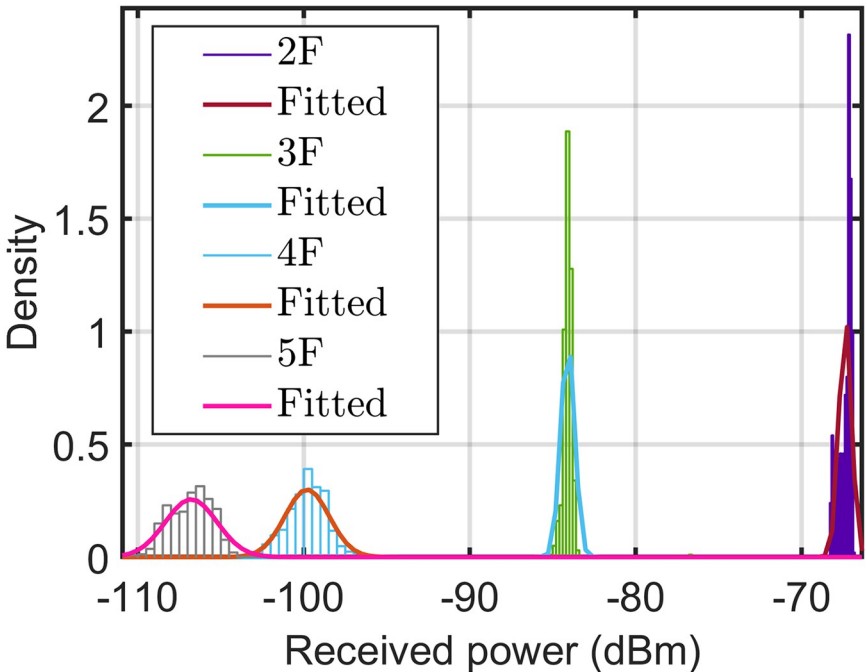

**Fig 6. Distribution of measured path loss along vertical Euclidean distance at 3.7 GHz using the horn-omnidirectional antenna.**

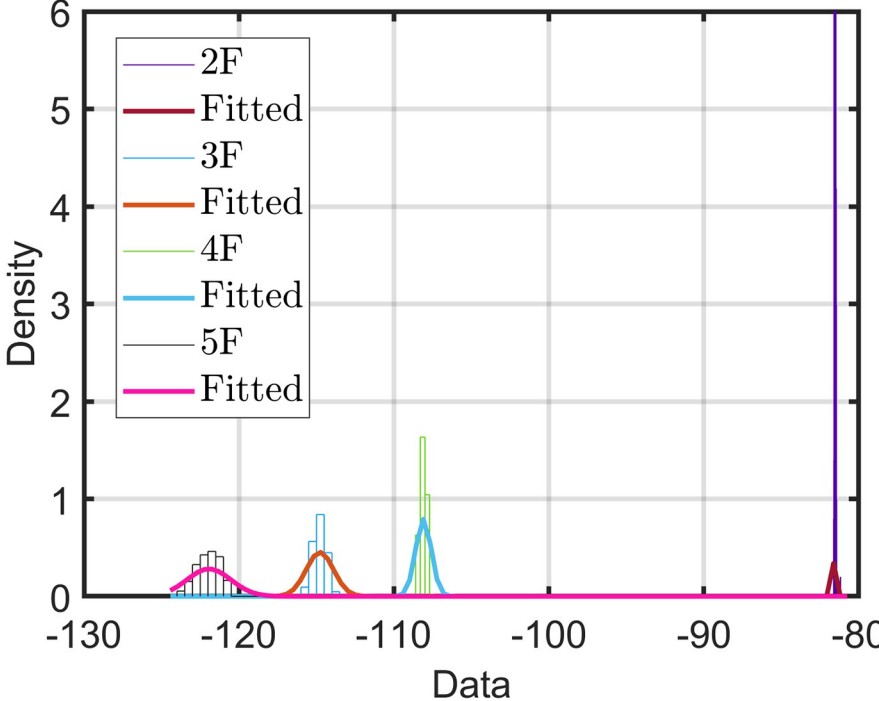

**Fig 7. Distribution of measured path loss along vertical Euclidean distance at 28 GHz using the horn-horn antenna.**

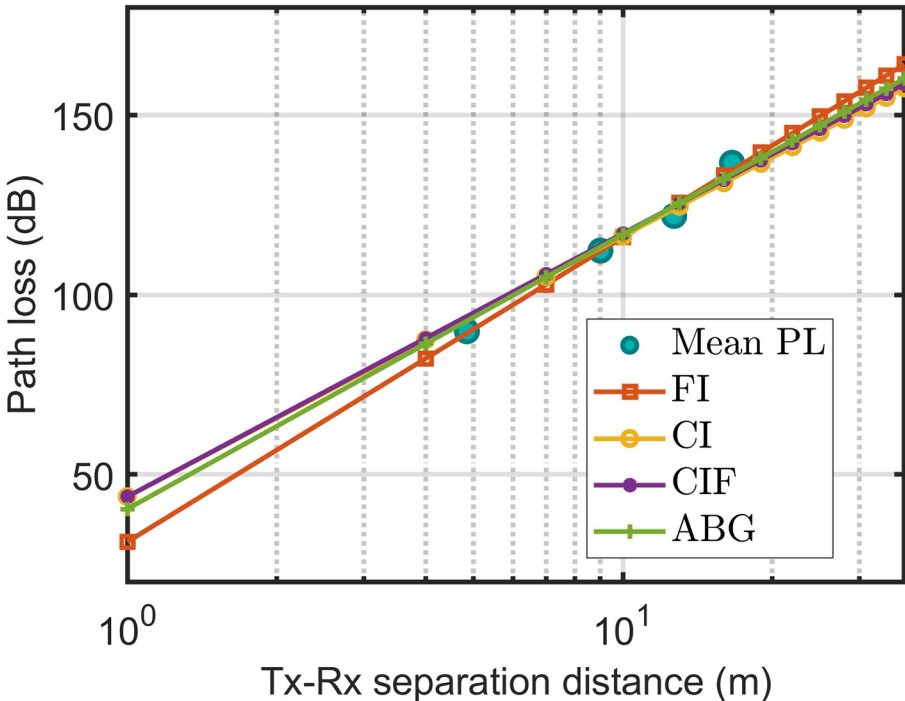

**Fig 8. The measured data fitted with path loss technique at 3.7 GHz H-H polarization and horn antenna.**

In Fig 8, the correlation between the average path loss and large-scale models indicates that each model was well-fitted with the average path loss. However, the ABM model fits the average path loss most appropriately among all the models.

Fig 9 indicates that for the omnidirectional antenna at 3.7 GHz, all models were well fitted with the measured path loss. Therefore, when precisely checking the shadow factor of the models, the standard deviations (of the shadow factor) were obtained as 1.32, 1.29, 1.45, and 1.57 for the CIDF, ABM, CIDMF, and ABGM models, respectively. Therefore, the ABM model was the best candidate among all the models.

Fig 10 shows that CIDF and CIDMF exhibited almost the same performance; therefore, the lines of CIDF and CIDMF overlapped. There were no significant differences between the CIDF, CIDMF, ABM, and ABGM models in terms of accuracy in the range of 5 to 20 m. However, beyond this range, slight deviations were observed. Therefore, when precisely checking the shadow factor of the models in Table 5, the standard deviations of the shadow factor were obtained as 6.63, 6.48, 6.63, and 6.63 for the CIDF, ABM, CIDMF, and ABGM models, respectively. Therefore, the ABM model was the best candidate among all the models.

Tables 6 and 7 (Table 7 is the extended part of Table 6) present the statistical characteristics of large-scale models, associated PLEs, and shadow factors from various research work. Although the geometry of the stairs, the measurement set ups, and the analysis performed vary between studies, the results frequently follow the same trends. In our study, for the 3.7 horn, 3.7 GHz omnidirectional, and 28 GHz horn antenna links, the PLEs were 7.28, 7.38, and 7.62, respectively, for the CI model. These PLE values are comparable to those found in [15] as 7.95 at 2.4 GHz, 8.13 at 5.8 GHz, and in [22] as 7.33 (site 1) at 28 GHz, 6.99 (site) at 28 GHz with the same H-H polarization. In [21], the reported PLEs are 7.90 at V-V polarization, comparable to our result of 7.62 at a frequency of 28 GHz for H-H polarization. However, the PLEs

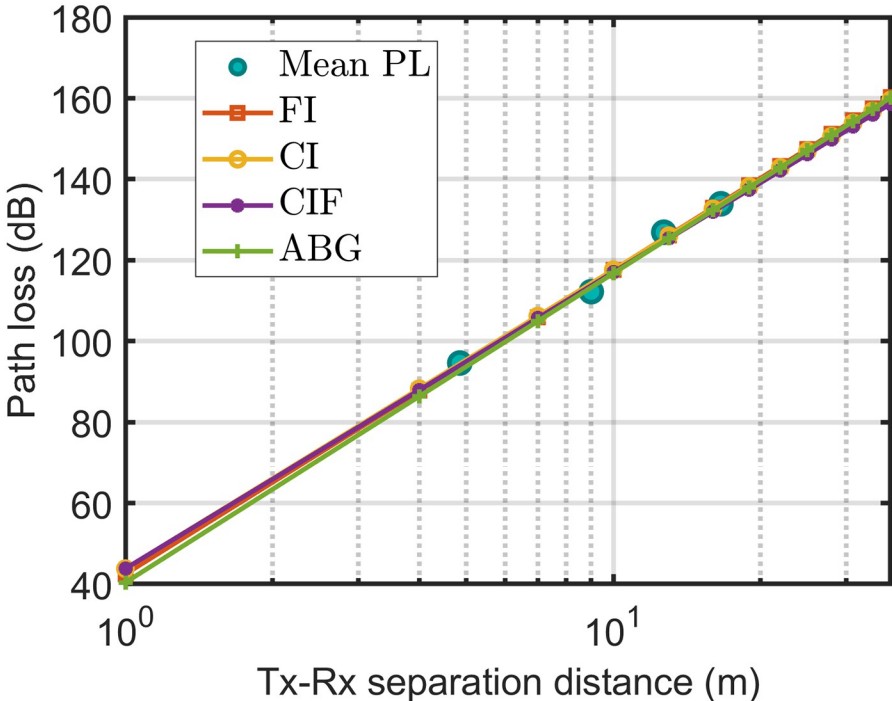

**Fig 9. The measured data fitted with path loss technique at 3.7 GHz H-H polarization and omnidirectional antenna.**

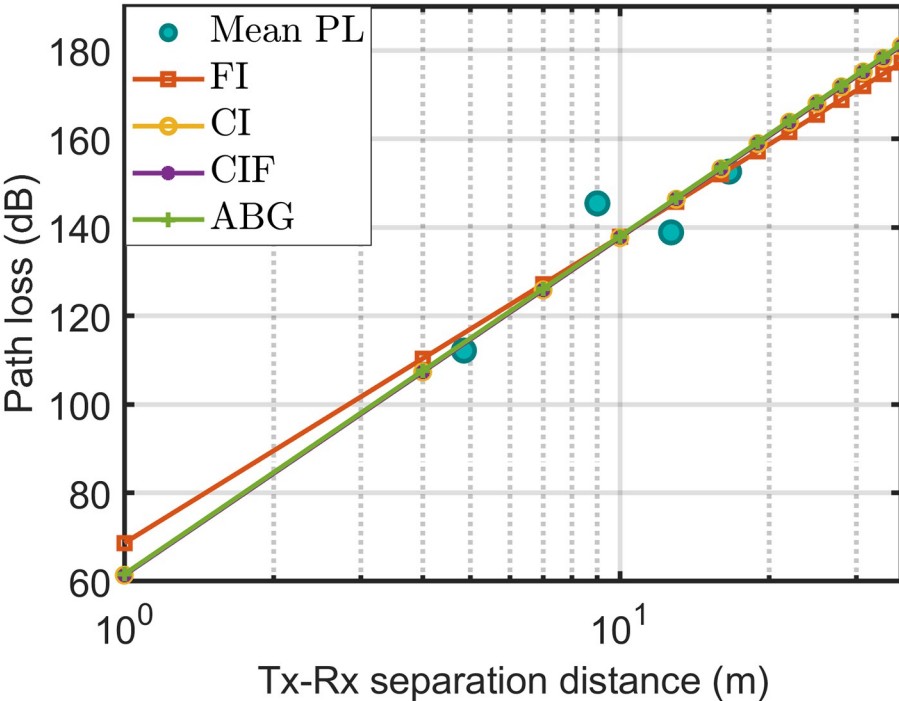

**Fig 10. The measured data fitted with path loss technique at 28 GHz H-H polarization and horn antenna.**

**Table 5. Path loss coefficients of path loss models at the stairwell (f in GHz).**

| f | FSPL | | | | PLE | | | | $b, \gamma$ | | STD (dB) | | | |
|---|------|------|-------|------|------|------|-------|------|-------|------|------|------|-------|------|
|   | CIDF | ABM | CIDMF | ABGM | CIDF | ABM | CIDMF | ABGM | CIDMF | ABGM | CI | ABM | CIDMF | ABGM |
| 3.7 | 43.81 | 31.25 | 43.81 | 26.66 | 7.28 | 8.47 | 7.42 | 7.63 | 0.020 | 2.41 | 3.04 | 1.76 | 3.12 | 2.57 |
| 3.7 | 43.81 | 42.47 | 43.81 | | 7.38 | 7.51 | | | | | 1.32 | 1.29 | 1.45 | 1.57 |
| 28 | 61.38 | 68.64 | 61.38 | | 7.62 | 6.92 | | | | | 6.63 | 6.48 | 6.63 | 6.63 |

reported in [23] are 2.00 and 1.90, which are much lower than all other studies, as shown in the second column of Table 7, including this study. The lower value of PLE may be because, in that study, the authors used omnidirectional-omnidirectional (V-V) and omnidirectional-horn (V-V) antennas, which are very different from the study in [15, 21, 22, 33].

The configuration of the reported study was similar to our research regarding the operating frequency and the same polarized links. Considering the polarization and operational frequency band similarity, our 28 GHz H-H band operation and that in [22] had similarities. However, these two studies have dissimilarities considering the link-type LOS and NLOS. Compared with this study, the studies reported in Tables 6 and 7 have a higher level of dissimilarities. However, later, we will see that the statistical parameters reported from the path loss model [23] were somewhat different, which may be because of the dissimilarity in the antenna configuration used.

In our study, in the case of the ABM models, the PLEs were obtained as 8.47, 7.51, and 6.92 for the 3.7 horn, the 3.7 GHz omnidirectional, and the 28 GHz horn antenna links, respectively; which are also comparable with the PLEs of 9.80 and 9.20 at the 28 GHz V-V link [21], 9.86 at the 28 GHz H-H link (site 1) [22], and 8.97 at the 28 GHz H-H link (site 2) [22]. However, the PLEs reported in [23] are much lower even for the ABM model, which is not comparable to the results in Table 7.

The average of these PLEs was 7.42, which is comparable to the average reported PLEs of 6.10 [21], 7.18, and 7.44, for sites 1 and 2 [21, 22], respectively, although the polarization configurations are different.

Our study obtained a PLE of 7.63 for the ABGM model with V-V polarization links. Other studies also reported comparable PLEs for the ABGM model despite polarization variations in 8.22–9.80 [21, 22].

The standard deviation of path loss of the CIDF, ABM, CIDMF, and ABGM models and the same parameter values in other reported studies; others reported a comparatively higher standard deviation value at all experimental sites [15, 21, 22]. However, in certain cases, our reported values were similar to those reported in a recent study [23, 33]. Comparing the standard deviation within our results, it is evident that the minimum standard deviation values were available among the 3.7 and 28 GHz antenna links in the omnidirectional configuration. These results were comparable to the lower standard deviation values reported for omnidirectional antenna links.

The second minimum standard deviation was offered in the 3.7 GHz operation link for all four models. The standard deviation values provided by the 28 GHz link were the highest among all the standard deviation values. Thus, the above discussions justify the validity of the measured data with the current research in stairwells.

## Conclusion

This study proposed path loss models with varying distances and frequencies in an indoor staircase based on observations in wall-separated stairwells with different antenna

**Table 6. Related works are compared mmWave propagation at stairwells.**

| Ref. | Freq. | P-P | Link Type | Antenna Type Tx-Rx | $f_0, d_0$ | | FSPL (dB) | | | |
|------|-------|-----|-----------|---------------------|------|------|------|------|------|------|
| | | | | | $f_0$ | $d_0$ | CIDF | $\alpha_{ABM}$ | CIDMF | $\beta_{ABGM}$ |
| [21] | 26 | V-V | NLOS | Horn-Horn | 38.00 | 5.50 | 60.70 | 19.10 | 60.70 | 99.00 |
| | 28 | V-V | NLOS | Horn-Horn | | | 61.34 | 28.00 | 61.34 | |
| | 32 | V-V | NLOS | Horn-Horn | | | 62.50 | 33.80 | 62.50 | |
| | 38 | V-V | NLOS | Horn-Horn | | | 64.00 | 30.30 | 64.00 | |
| Site 1 [22] | 26 | V-V | NLOS/LOS | Horn-Horn | 31.00 | 1.00 | 60.70 | – | 60.70 | 12.90 |
| | 26 | H-H | NLOS/LOS | Horn-Horn | | | 60.70 | | 60.70 | |
| | 26 | V-H | NLOS/LOS | Horn-Horn | | | 60.70 | | 60.70 | |
| | 28 | V-V | NLOS/LOS | Horn-Horn | 31.00 | 1.00 | 61.34 | – | 61.34 | 12.90 |
| | 28 | H-H | NLOS/LOS | Horn-Horn | | | 61.34 | | 61.34 | |
| | 28 | V-H | NLOS/LOS | Horn-Horn | | | 61.34 | | 61.34 | |
| | 32 | V-V | NLOS/LOS | Horn-Horn | 31.00 | 1.00 | 62.50 | – | 62.50 | 12.90 |
| | 32 | H-H | NLOS/LOS | Horn-Horn | | | 62.50 | | 62.50 | |
| | 32 | V-H | NLOS/LOS | Horn-Horn | | | 62.50 | | 62.50 | |
| | 38 | V-V | NLOS/LOS | Horn-Horn | 31.00 | 1.00 | 64.00 | – | 64.00 | 12.90 |
| | 38 | H-H | NLOS/LOS | Horn-Horn | | | 64.00 | | 64.00 | |
| | 38 | V-H | NLOS/LOS | Horn-Horn | | | 64.00 | | 64.00 | |
| Site 2 [22] | 26 | V-V | NLOS/LOS | Horn-Horn | 31.00 | 1.00 | 60.70 | – | 60.70 | 8.81 |
| | 26 | H-H | NLOS/LOS | Horn-Horn | | | 60.70 | | 60.70 | |
| | 26 | V-H | NLOS/LOS | Horn-Horn | | | 60.70 | | 60.70 | |
| | 28 | V-V | NLOS/LOS | Horn-Horn | 31.00 | 1.00 | 61.34 | – | 61.34 | 8.81 |
| | 28 | H-H | NLOS/LOS | Horn-Horn | | | 61.34 | | 61.34 | |
| | 28 | V-H | NLOS/LOS | Horn-Horn | | | 61.34 | | 61.34 | |
| | 32 | V-V | NLOS/LOS | Horn-Horn | 31.00 | 1.00 | 62.50 | – | 62.50 | 8.81 |
| | 32 | H-H | NLOS/LOS | Horn-Horn | | | 62.50 | | 62.50 | |
| | 32 | V-H | NLOS/LOS | Horn-Horn | | | 62.50 | | 62.50 | |
| | 38 | V-V | NLOS/LOS | Horn-Horn | 31.00 | 1.00 | 64.00 | – | 64.00 | 8.81 |
| | 38 | H-H | NLOS/LOS | Horn-Horn | | | 64.00 | | 64.00 | |
| | 38 | V-H | NLOS/LOS | Horn-Horn | | | 64.00 | | 64.00 | |
| [15] | 2.4 | V-V | NLOS/LOS | Dipole-Dipole | – | – | 40.00 | – | – | – |
| | 2.4 | H-H | NLOS/LOS | Dipole-Dipole | – | – | 40.00 | – | – | – |
| | 5.8 | V-V | NLOS/LOS | Dipole-Dipole | – | – | 47.67 | – | – | – |
| | 5.8 | H-H | NLOS/LOS | Dipole-Dipole | – | – | 47.67 | – | – | – |
| [23] | 3.5 | V-V | NLOS/LOS | Omni.-Omni. | 16.00 | 0.50 | 43.28 | 36.60 | 43.28 | – |
| | 28 | V-V | NLOS/LOS | Omni.-Horn | 16.00 | 0.50 | 61.34 | 57.30 | 61.34 | – |
| [33] | 26 | V-V | NLOS | Omni. biconical-Horn | – | – | – | – | – | – |
| | | V-H | NLOS | Omni. biconical-Horn | – | – | – | – | – | – |
| | 38 | V-V | NLOS | Omni. biconical-Horn | – | – | – | – | – | – |
| | | V-H | NLOS | Omni. biconical-Horn | – | – | – | – | – | – |
| | 26 | V-V | NLOS | Omni. biconical-Horn | – | – | – | – | – | – |
| | | V-H | NLOS | Omni. biconical-Horn | – | – | – | – | – | – |
| | 38 | V-V | NLOS | Omni. biconical-Horn | – | – | – | – | – | – |
| | | V-H | NLOS | Omni. biconical-Horn | – | – | – | – | – | – |
| This study | 3.7 | H-H | NLOS | Horn-Horn | 6.13 | 1.00 | 43.76 | 31.25 | 43.76 | 26.66 |
| | 3.7 | H-H | NLOS | Horn-Omni. | 6.13 | 1.00 | 43.76 | 42.47 | 43.76 | 26.66 |
| | 28 | H-H | NLOS | Horn-Horn | 6.13 | 1.00 | 61.34 | 68.64 | 61.34 | 26.66 |

**Table 7. Related works are compared mmWave propagation at stairwells.**

| Ref. | $n$ | | | | $b, \gamma$ | | Standard deviation (dB) | | | |
|---|---|---|---|---|---|---|---|---|---|---|
| | $n_{CIDF}$ | $\beta_{ABM}$ | $n_{CIDMF}$ | $\alpha$ | $b$ | $\gamma_{ABGM}$ | $\sigma_{CIDF}$ | $\sigma_{ABM}$ | $\sigma_{CIDMF}$ | $\sigma_{ABGM}$ |
| [21] | 7.40 | 10.90 | 6.10 | 9.80 | -0.50 | -4.80 | 4.40 | 6.30 | 10.10 | 7.40 |
| | 7.90 | 9.80 | | | | | 5.10 | 6.70 | | |
| | 6.60 | 9.20 | | | | | 5.10 | 7.60 | | |
| | 7.10 | 9.10 | | | | | 6.40 | 8.10 | | |
| Site 1 [22] | 7.52 | 10.92 | 7.44 | 9.75 | -0.04 | 1.57 | 9.95 | 6.28 | 9.04 | 7.38 |
| | 7.31 | | | | | | 9.00 | | | |
| | 7.79 | | | | | | 6.93 | | | |
| | 7.60 | 9.86 | 7.44 | 9.75 | -0.04 | 1.57 | 8.86 | 6.67 | 9.04 | 7.38 |
| | 7.33 | | | | | | 8.99 | | | |
| | 7.86 | | | | | | 5.57 | | | |
| | 7.48 | 9.17 | 7.44 | 9.75 | -0.04 | 1.57 | 8.49 | 7.58 | 9.04 | 7.38 |
| | 7.60 | | | | | | 8.65 | | | |
| | 7.96 | | | | | | 7.89 | | | |
| | 7.36 | 9.09 | 7.44 | 9.75 | -0.04 | 1.57 | 8.98 | 8.07 | 9.04 | 7.38 |
| | 7.26 | | | | | | 8.95 | | | |
| | 7.57 | | | | | | 5.55 | | | |
| Site 2 [22] | 7.06 | 9.81 | 7.18 | 8.22 | -0.01 | 2.81 | 10.26 | 8.01 | 10.34 | 10.06 |
| | 7.36 | | | | | | 12.10 | | | |
| | 7.52 | | | | | | 8.60 | | | |
| | 7.15 | 8.97 | 7.18 | 8.22 | -0.01 | 2.81 | 10.46 | 9.60 | 10.34 | 10.06 |
| | 6.99 | | | | | | 13.55 | | | |
| | 7.27 | | | | | | 6.48 | | | |
| | 7.55 | 7.37 | 7.18 | 8.22 | -0.01 | 2.81 | 9.98 | 9.98 | 10.34 | 10.06 |
| | 7.35 | | | | | | 11.51 | | | |
| | 7.57 | | | | | | 7.56 | | | |
| | 6.99 | 6.73 | 7.18 | 8.22 | -0.01 | 2.81 | 9.69 | 9.67 | 10.34 | 10.06 |
| | 6.99 | | | | | | 10.45 | | | |
| | 7.29 | | | | | | 8.52 | | | |
| [15] | 8.88 | – | – | – | – | – | 5.72 | – | – | – |
| | 7.95 | – | – | – | – | – | 4.67 | – | – | – |
| | 11.34 | – | – | – | – | – | 6.23 | – | – | – |
| | 8.13 | – | – | – | – | – | 2.53 | – | – | – |
| [23] | 2.00 | 1.90 | 1.90 | – | 0.05 | – | 0.50 | 0.40 | 0.90 | – |
| | 1.90 | 1.60 | 1.90 | – | 0.05 | – | 0.50 | 0.30 | 0.90 | – |
| [33] | 4.18 | – | – | – | – | – | 2.11 | – | – | – |
| | 4.46 | – | – | – | – | – | 10.90 | – | – | – |
| | 3.97 | – | – | – | – | – | 8.61 | – | – | – |
| | 4.00 | – | – | – | – | – | 10.70 | – | – | – |
| | 2.31 | – | – | – | – | – | 2.34 | – | – | – |
| | 2.31 | – | – | – | – | – | 1.66 | – | – | – |
| | 1.65 | – | – | – | – | – | 0.96 | – | – | – |
| | 1.65 | – | – | – | – | – | 1.51 | – | – | – |
| This study | 7.28 | 8.47 | 7.42 | 7.63 | 0.02 | 2.41 | 3.04 | 1.76 | 3.12 | 2.57 |
| | 7.38 | 7.51 | 7.42 | 7.63 | 0.02 | 2.41 | 1.32 | 1.29 | 1.45 | 1.57 |
| | 7.62 | 6.92 | 7.42 | 7.63 | 0.02 | 2.41 | 6.63 | 6.48 | 6.63 | 6.63 |

polarizations. We also extracted the path loss exponents and standard deviations from the recorded datasets associated with the measured path losses. The findings were compared with previous studies regarding polarization in the transmitter and receiver, frequency ranges, and stairwell configurations. The comparative discussion verified the statistics of the measured data with the previously measured datasets. All four models fit well with the measured data and show good compatibility. However, among the four models, the ABM model exhibited a comparatively lower standard deviation, implying that the probability of path loss variations is lower for the ABM model.

## Supporting information

**S1 File.**
(PDF)

**S1 Data.**
(XLSX)

**S2 Data.**
(XLSX)

**S3 Data.**
(XLSX)

**S1 Appendix.**
(PDF)

## Author Contributions

**Conceptualization:** Md Abdus Samad, Dong-You Choi.

**Data curation:** Md Abdus Samad.

**Formal analysis:** Md Abdus Samad.

**Funding acquisition:** Kwonhue Choi.

**Investigation:** Md Abdus Samad.

**Methodology:** Md Abdus Samad.

**Project administration:** Kwonhue Choi.

**Resources:** Md Abdus Samad, Dong-You Choi, Kwonhue Choi.

**Software:** Md Abdus Samad.

**Supervision:** Dong-You Choi, Kwonhue Choi.

**Validation:** Md Abdus Samad.

**Visualization:** Md Abdus Samad.

**Writing – original draft:** Md Abdus Samad.

**Writing – review & editing:** Md Abdus Samad, Dong-You Choi, Kwonhue Choi.

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
