## [Decision Letter · Decision Letter 0]

8 Dec 2022

PONE-D-22-28055Path Loss Measurement and Modeling of 5G Network in Emergency Indoor StairwellPLOS ONE

Dear Dr. Samad,

Thank you for submitting your manuscript to PLOS ONE. After careful consideration, we feel that it has merit but does not fully meet PLOS ONE’s publication criteria as it currently stands. Therefore, we invite you to submit a revised version of the manuscript that addresses the points raised during the review process.

We look forward to receiving your revised manuscript.

Kind regards,

Chan Hwang See, Ph.D.

Academic Editor

PLOS ONE

Journal Requirements:

"This work was supported  by the National Research Foundation of Korea (NRF) grant funded by the Korean government (MSIT) under Grant 2021R1A2C1010370."

"This work was supported  by the National Research Foundation of Korea (NRF) grant funded by the Korean government (MSIT) under Grant 2021R1A2C1010370."

Reviewers' comments:

Reviewer's Responses to Questions

**Comments to the Author**

1. Is the manuscript technically sound, and do the data support the conclusions?

Reviewer #1: Partly

Reviewer #2: No

Reviewer #3: Yes

2. Has the statistical analysis been performed appropriately and rigorously? 

Reviewer #1: Yes

Reviewer #2: No

Reviewer #3: Yes

3. Have the authors made all data underlying the findings in their manuscript fully available?

Reviewer #1: Yes

Reviewer #2: No

Reviewer #3: Yes

4. Is the manuscript presented in an intelligible fashion and written in standard English?

Reviewer #1: Yes

Reviewer #2: No

Reviewer #3: Yes

5. Review Comments to the Author

Reviewer #1: The paper covers a timely and interesting topic. This reviewer has the below comments.

1. The authors should provide a table summarizing the main notation that has been adopted in the manuscript.

2. Authors need to cover a few operational frequencies to make the work more robust and prove the accuracy of the work. If it is not possible due to measurement data then should refer to the operation frequency in the title.

3. Need to present some clear modelling layouts where signal details information should be found. For more details, the authors can read some recent works regarding radio propagation modelling such as (https://doi.org/10.1016/j.aej.2022.04.033, https://doi.org/10.1371/journal.pone.0201905, and https://doi.org/10.3390/electronics8030286 ).

4. References are not up to date. Need to align with the literature on indoor radio propagation research.

5. Proposed method detail is missing. Need to demonstrate the proposed method performance with respect of any other benchmark method .

6. Simulation details information needed including simulation configuration parameters such as Antenna gain, cable loss etc.

Reviewer #2: This is a very difficult paper to review. It needs to put the images embedded in the text where they occur. This is obviously the best thing to do as it is not at all easy scrolling down to an image from the text and being unsure which one it is referring to.

Here an empirical propagation model has been measured for propagation through stair wells. This should be done for different locations to show consistency/inconsistency. But then is measuring the stairwell only of any relevance? Rather what is needed is to have a horn antenna illuminate the bottom of the stairwell to represent the extra loss likely to be found from any EM waves entering the stairwell from the floor. Then the antenna up the stairwell would be the mobile and thus should be omnidirectional. Really what is needed is samples of measurements up the stairs vs height, so that then it can be marked clearly where each floor is and then some idea be given as to how the propagation works and if path loss exponents do change. This is not clear from the data and certainly only data need be presented as path losses, not power received.

For mmWave this is not so useful information given it is NLOS and does not really work well at that band up stairs with line of sight blockage. With the need for directive antennas it is hard to determine the max position where there is loss or has it been averaged based on taking all azimuth directions? The elevation would likewise also matter there.

This paper certainly can do with more clear documentaton of results that show the real actual path loss and not power levels as well as ensuring the interpretation of them.

Reviewer #3: The paper is good and technically sound. The measurements are designed and presented in a good way. However, two minor issues need to be revised in order to improve the paper quality

1- the abstract is not consice enough and need to be rewritten in better way to give more insights about the motivation and contributions

2- More dissuction about the motivation and research gap should be added in the introduction.

6. PLOS authors have the option to publish the peer review history of their article (what does this mean?). If published, this will include your full peer review and any attached files.

Reviewer #1: **Yes: **Dr. Ferdous Hossain

Reviewer #2: No

Reviewer #3: **Yes: **Abdulmajid Al-Mqdashi

---

## [Author Response · Author response to Decision Letter 0]

14 Jan 2023

Response to Reviewers

Reviewer #1: The paper covers a timely and interesting topic. This reviewer has the below comments.

1. The Author’s should provide a table summarizing the main notation that has been adopted in the manuscript.

Author’s response: We have the summarizing table containing all the symbols used in Table 8.

2. Author’s need to cover a few operational frequencies to make the work more robust and prove the accuracy of the work. If it is not possible due to measurement data then should refer to the operation frequency in the title.

Author’s response: We have used the operating frequency in the manuscript title.

3. Need to present some clear modelling layouts where signal details information should be found. For more details, the Author’s can read some recent works regarding radio propagation modelling such as (https://doi.org/10.1016/j.aej.2022.04.033, https://doi.org/10.1371/journal.pone.0201905, and https://doi.org/10.3390/electronics8030286 ).

Author’s response: We have added Tables 1, 2, and 3 and procedure to calculate path loss from the received power in lines 159—173.

4. References are not up to date. Need to align with the literature on indoor radio propagation research.

Author’s response: We have revised the introduction and background study by adding related indoor radio propagation research work. (lines 4—22)

5. Proposed method detail is missing. Need to demonstrate the proposed method performance with respect of any other benchmark method .

Author’s response: We have already compared our results with the available results in the literature about the path measurement result in the emergency stair in Table 6 and 7.

6. Simulation details information needed including simulation configuration parameters such as Antenna gain, cable loss etc.

Author’s response: We have given the real measurement configuration parameters in this version of the manuscript (Tables 1, 2, and 3).

Reviewer #2: This is a very difficult paper to review. It needs to put the images embedded in the text where they occur. This is obviously the best thing to do as it is not at all easy scrolling down to an image from the text and being unsure which one it is referring to.

Author’s response: According to the format of this journal, we cannot insert images inside the text—the current style of submission of this journal during the review process.

Here an empirical propagation model has been measured for propagation through stair wells. This should be done for different locations to show consistency/inconsistency. But then is measuring the stairwell only of any relevance? Rather what is needed is to have a horn antenna illuminate the bottom of the stairwell to represent the extra loss likely to be found from any EM waves entering the stairwell from the floor. Then the antenna up the stairwell would be the mobile and thus should be omnidirectional. Really what is needed is samples of measurements up the stairs vs height, so that then it can be marked clearly where each floor is and then some idea be given as to how the propagation works and if path loss exponents do change. This is not clear from the data and certainly only data need be presented as path losses, not power received.

Author’s response: We have already analyzed the consistency/inconsistency of the measured data in table 4.

Please see other measurement scenarios in the literature, for example, (1) doi:10.3906/elk-1710-248, (2) doi:10.1109/LAWP.2019.2961641, about placing the antenna in the emergency stair.

We have added a description of how the path losses were calculated from the received power (in lines 159—173).

For mmWave this is not so useful information given it is NLOS and does not really work well at that band up stairs with line of sight blockage. With the need for directive antennas it is hard to determine the max position where there is loss or has it been averaged based on taking all azimuth directions? The elevation would likewise also matter there.

Author’s response: The literature uses different antenna configuration types in the studied wireless link. Various researchers used horn-to-horn, dipole-to-dipole, or omnidirectional-to-horn type antennae in the reported study (please see other research works as reported in Table 4.). In this study, we used the horn antenna in the transmitter and the antenna in the receiver in a special type of stair environment (stairs are separated by wall) that has not been studied before. 

This paper certainly can do with more clear documentaton of results that show the real actual path loss and not power levels as well as ensuring the interpretation of them.

Author’s response: We have already simulated the actual path loss in Figures 8, 9, and 10. In addition, calculated path losses are given in Tables 1, 2, and 3.

Reviewer #3: The paper is good and technically sound. The measurements are designed and presented in a good way. However, two minor issues need to be revised in order to improve the paper quality

1- the abstract is not consice enough and need to be rewritten in better way to give more insights about the motivation and contributions

Author’s response: We have revised the abstract.

2- More dissuction about the motivation and research gap should be added in the introduction.

Author’s response: We have revised the motivation and research gap in lines 5—22, 38—42, and 74—81.

---

## [Decision Letter · Decision Letter 1]

8 Feb 2023

PONE-D-22-28055R1Path Loss Measurement and Modeling of 5G Network in Emergency Indoor Stairwell at 3.7 and 28 GHzPLOS ONE

Dear Dr. Samad,

Thank you for submitting your manuscript to PLOS ONE. After careful consideration, we feel that it has merit but does not fully meet PLOS ONE’s publication criteria as it currently stands. Therefore, we invite you to submit a revised version of the manuscript that addresses the points raised during the review process.

We look forward to receiving your revised manuscript.

Kind regards,

Chan Hwang See, Ph.D.

Academic Editor

PLOS ONE

Journal Requirements:

Additional Editor Comments (if provided):

After the first round of major revision, authors have addressed majority of the comments. But, there are still minor issues need to be addressed. Therefore, minor correction decision is recommended.

Reviewers' comments:

Reviewer's Responses to Questions

**Comments to the Author**

1. If the authors have adequately addressed your comments raised in a previous round of review and you feel that this manuscript is now acceptable for publication, you may indicate that here to bypass the “Comments to the Author” section, enter your conflict of interest statement in the “Confidential to Editor” section, and submit your "Accept" recommendation.

Reviewer #2: All comments have been addressed

Reviewer #3: All comments have been addressed

2. Is the manuscript technically sound, and do the data support the conclusions?

Reviewer #2: Partly

Reviewer #3: Yes

3. Has the statistical analysis been performed appropriately and rigorously? 

Reviewer #2: Yes

Reviewer #3: Yes

4. Have the authors made all data underlying the findings in their manuscript fully available?

Reviewer #2: Yes

Reviewer #3: No

5. Is the manuscript presented in an intelligible fashion and written in standard English?

Reviewer #2: Yes

Reviewer #3: Yes

6. Review Comments to the Author

Reviewer #2: This paper is still very difficult to review and I don't agree with the journal requirements that the figures have to be not embedded in the text. It is a no brainer that it is far easier to read a paper with the figures in the right place. I still see no point in publishing the received power levels but at least the total path loss is shown in the final column. Also units dB and dBm need to be indicated in the headings of the tables or in the captions. The path loss exponents are eventually shown that are the important point of documentation in this work.

Subscripts as words like "loss" and acronyms like "FSPL" in the equations should not be italic. This all needs to be consistent.

Reviewer #3: The paper is revised well and improved by the authors which make it now qualified . So, I recommend it for publication

7. PLOS authors have the option to publish the peer review history of their article (what does this mean?). If published, this will include your full peer review and any attached files.

Reviewer #2: No

Reviewer #3: **Yes: **Abdulmajid Al-Mqdashi

---

## [Author Response · Author response to Decision Letter 1]

16 Feb 2023

Response to Reviewers

Reviewer #2

1. If the authors have adequately addressed your comments raised in a previous round of review and you feel that this manuscript is now acceptable for publication, you may indicate that here to bypass the “Comments to the Author” section, enter your conflict of interest statement in the “Confidential to Editor” section, and submit your "Accept" recommendation.

Reviewer #2: All comments have been addressed

Author’s response: We want to thank the reviewer.________________________________________

2. Is the manuscript technically sound, and do the data support the conclusions?

Reviewer #2: Partly

Author’s response: We want to thank the reviewer.

3. Has the statistical analysis been performed appropriately and rigorously?

Reviewer #2: Yes

Author’s response: We want to thank the reviewer.

4. Have the authors made all data underlying the findings in their manuscript fully available?

Reviewer #2: Yes

Author’s response: We want to thank the reviewer.

5. Is the manuscript presented in an intelligible fashion and written in standard English?

Reviewer #2: Yes

Author’s response: We want to thank the reviewer.

6. Review Comments to the Author

Reviewer #2: (a) This paper is still very difficult to review and I don't agree with the journal requirements that the figures have to be not embedded in the text. It is a no brainer that it is far easier to read a paper with the figures in the right place. (b) I still see no point in publishing the received power levels but at least the total path loss is shown in the final column. (c) Also units dB and dBm need to be indicated in the headings of the tables or in the captions. The path loss exponents are eventually shown that are the important point of documentation in this work.

(d) Subscripts as words like "loss" and acronyms like "FSPL" in the equations should not be italic. This all needs to be consistent.

Author’s response: (a) We have nothing to do as the format set by the journal. (b) We have removed the path loss in Table 1, 2, and 3. (c) We have added power level measured in dB in Table 4, 5, and 6. (d) We have made the mentioned symbols, and other subscripts such as ABG, ABGM, CIDF, and CIDMF in equations (1)—(6), in the text below of these equations, and also in Table 8. 

Reviewer #3

1. If the authors have adequately addressed your comments raised in a previous round of review and you feel that this manuscript is now acceptable for publication, you may indicate that here to bypass the “Comments to the Author” section, enter your conflict of interest statement in the “Confidential to Editor” section, and submit your "Accept" recommendation.

Reviewer #3: All comments have been addressed

Author’s response: We want to thank the reviewer.

2. Is the manuscript technically sound, and do the data support the conclusions?

Reviewer #3: Yes

Author’s response: We want to thank the reviewer.

3. Has the statistical analysis been performed appropriately and rigorously?

Reviewer #3: Yes

Author’s response: We want to thank the reviewer.

4. Have the authors made all data underlying the findings in their manuscript fully available?

Reviewer #3: No

Author’s response: We have already added the measured datasets in Table 1, 2, and 3.

5. Is the manuscript presented in an intelligible fashion and written in standard English?

Reviewer #3: Yes

Author’s response: We want to thank the reviewer.

6. Review Comments to the Author

Reviewer #3: The paper is revised well and improved by the authors which make it now qualified . So, I recommend it for publication

Author’s response: We want to thank the reviewer.

---

## [Editor Report · Decision Letter 2]

23 Feb 2023

Path Loss Measurement and Modeling of 5G Network in Emergency Indoor Stairwell at 3.7 and 28 GHz

PONE-D-22-28055R2

Dear Dr. Samad,

We’re pleased to inform you that your manuscript has been judged scientifically suitable for publication and will be formally accepted for publication once it meets all outstanding technical requirements.

Kind regards,

Chan Hwang See, Ph.D.

Academic Editor

PLOS ONE

Additional Editor Comments (optional):

Authors have addressed all the comments from reviewers. Therefore, this paper is recommended to publish in this journal.
---

## [Editor Report · Acceptance letter]

2 Mar 2023

PONE-D-22-28055R2 

Path Loss Measurement and Modeling of 5G Network in Emergency Indoor Stairwell at 3.7 and 28 GHz 

Dear Dr. Samad:

I'm pleased to inform you that your manuscript has been deemed suitable for publication in PLOS ONE. Congratulations! Your manuscript is now with our production department. 

Kind regards, 

on behalf of

Dr. Chan Hwang See 

Academic Editor

PLOS ONE